# Surface Modification of Okara Cellulose Crystals with Phenolic Acids to Prepare Multifunction Emulsifier with Antioxidant Capacity and Lipolysis Retardation Effect

**DOI:** 10.3390/foods13020184

**Published:** 2024-01-05

**Authors:** Nopparat Prabsangob, Sasithorn Hangsalad, Sunsanee Udomrati

**Affiliations:** 1Department of Product Development, Faculty of Agro-Industry, Kasetsart University, Bangkok 10900, Thailand; 2Department of Food Chemistry and Physics, Institute of Food Research and Product Development, Kasetsart University, Bangkok 10900, Thailand

**Keywords:** emulsifier, cellulose crystal, lipolysis retardation, phenolics, antioxidant activity, surface modification

## Abstract

Emulsion-based foods are widely consumed, and their characteristics involving colloidal and oxidative stabilities should be considered. The fabrication of the interfaces by selecting the emulsifier may improve stability and trigger lipolysis, thereby reducing energy uptake from the emulsified food. The present work aimed to develop Okara cellulose crystals (OCs) as a multifunction emulsifier to preserve the physical and chemical stability of a Pickering emulsion via surface modification with phenolic acids. The modification of OC was performed by grafting with the selected phenolics to produce OC–gallic acid (OC-G) and OC–tannic acid (OC-T) complexes. There was a higher phenolic loading efficiency when the OC reacted with gallic acid (ca. 70%) than with tannic acid (ca. 50%). This trend was concomitant with better antioxidant activity of the OC-G than OC-T. Surface modification based on grafting with phenolic acids improved capability of the OC to enhance both the colloidal and oxidative stability of the emulsion. In addition, the cellulosic materials had a retardation effect on the in vitro lipolysis compared to a protein-stabilized emulsion. Surface modification by grafting with phenolic acids successfully provided OC as an innovative emulsifier to promote physico-chemical stability and lower lipolysis of the emulsion.

## 1. Introduction

An oil-in-water (O/W) emulsion is found in several food products, such as milk, sauces, and mayonnaise. Emulsifiers are generally added to enhance the formability and dispersibility of the emulsion. Recently, Pickering emulsions, defined as emulsions stabilized using solid particles, have been gaining interest owing to their reduced requirement for a surfactant. A Pickering emulsion always has high stability due to an irreversible adsorption of the solid particles at interfacial areas. Furthermore, a Pickering emulsion has a practical role as an excipient for bioactive compounds in the development of functional foods [1]. For Pickering emulsion-based foods, proteins and polysaccharides are typically used as emulsifiers such as whey protein [2] and cellulosic materials [3]. Cellulose crystals (cellulosic materials with the amorphous region removed using a chemical or physical process or both) are interesting potential emulsifiers for a Pickering emulsion because their surface activity can effectively enhance the formability and stability of the emulsion system. Cellulose crystals, with suitable particle characteristics, could promote colloidal stability through both electrostatic and steric stabilization mechanisms [3]. With their capability to intermingle with other compounds involving the digestive enzymes and bile salts, cellulosic materials could alter lipolysis rate in the human GIT, resulting in lowering energy uptake from emulsified food consumption [4]. Unlike proteins, moreover, cellulosic materials cannot be digested by the human gastrointestinal tract (GIT), promoting their role as a dietary fiber with health-promoting effects [4].

An emulsion is typically prone to lipid oxidation due to a large contact area between the oil and the pro-oxidants present in the continuous phase, such as metal ions. Phenolic compounds are natural agents that are promising candidates to replace synthetic antioxidants in food production. In addition, phenolic compounds can exert bioactivity properties, such as the prevention of free radical formation and the accumulation of oxidized triglycerides [5]. However, phenolics are sensitive to food processing and storage conditions, such as high temperature and the presence of light and oxygen [6]. The susceptibility of phenolics against upper GIT activity also diminishes their bioaccessibility, resulting in reduced effectiveness of the bioactivities of the compounds [7].

Immobilization of phenolics with biopolymers is an interesting way to improve the stability and bioaccessibility of phenolic compounds, making this a promising way to integrate natural antioxidants into food product. Improvement of the thermal stability of phenolic compounds could be achieved via grafting with cellulosic materials [8]. Modification of biopolymers by grafting with phenolic compounds could improve functional properties of the biopolymers, while integration of carboxymethyl cellulose with sesame phenolic extracts provided cellulose-based films with efficient antioxidant and antimicrobial activities [9]. Enhancement of the heat stability of cellulose nanocrystals could be performed via surface modification with phenolic-rich Andean berry extract [6]. Most of these studies that considered the modification of celluloses using phenolic compounds focused on the production of packaging materials. Nevertheless, the use of modified cellulose-based materials as emulsifiers is still limited. Current studies on the improvement of the emulsifying ability of cellulose crystals to enhance the colloidal and oxidative stability of a Pickering emulsion have carried out investigations via surface modification with phenolic compounds. In this work, gallic acid and tannic acid were used as representative phenolic compounds according to their different molecular sizes. These selected phenolic compounds can be practically employed for food production owing to their safety and availability at low cost. Moreover, several health-promoting effects of gallic acid and tannic acid have been reported, such as antioxidant, antimutagenic, anticarcinogenic and blood lipid lowering capacities [5].

Okara, the by-product from soy-based food production, can be used to prepare okara cellulose crystal (OC) with potent emulsifying capability [3]. Presently, the development of OC as a multifunction emulsifier to preserve the physical and chemical stability of a Pickering emulsion has been performed via surface modification with phenolic acids. Furthermore, in vitro antioxidant and lipolysis retardation effects of the modified OC in the stimulated GIT have been studied.

## 2. Materials and Methods

### 2.1. Materials

Soybean oil was purchased from a local market (Bangkok, Thailand). Gallic acid, tannic acid, *1*,*1*-diphenyl-*2*-picrylhydrazyl (DPPH), *2*,*2*′-azinobis (*3*-ethylbenzothiazoline-*6*-sulfonic acid), diammonium salt (ABTS), *6*-hydroxy-*2*,*5*,*7*,*8*-tetramethylchromane-*2*-carboxylic acid (Trolox) and thiobarbituric acid were purchased (Sigma-Aldrich Chemical Co., St. Louis, MO, USA). The digestive components involving α-amylase from porcine pancreas Type VI-B (≥5 U/mg), pepsins from porcine gastric mucosa (≥250 U/mg), lipase from porcine pancreas (100–650 U/mg protein using olive oil as substrate) and bile extract from porcine were the products of Sigma-Aldrich Chemical Co (St. Louis, MO, USA). All used reagents were of analytical grade.

OC was prepared following the method of Prabsangob [3]. Briefly, the dried okara was initially purified using a ratio of hexane-to-ethanol (2:1, *v*/*v*) and aqueous NaOH solution (100 g/L) at a ratio of 1:5 (weight per volume, *w*/*v*) at ambient temperature for 7 h, before washing with excess water and then drying (60 °C, 12 h). The treated okara was reacted with a mixture of acetic acid (80%) and nitric acid (65%) at the ratio 1:5 (*w*/*v*) at 45 °C for 45 min to separate cellulose. Then, the cellulose was hydrolyzed with the aid of sulfuric acid (64.5%) at the ratio 1:10 (*w*/*v*) at 45 °C for 45 min. After the hydrolysis, the OC was washed with DI water and centrifuged several times, before dialyzing against DI water until the pH of the suspension was close to neutral, after which it was dried (60 °C, 12 h). The OC had a diameter of 0.75 μm, surface charge of −34.7 mV and a crystallinity index of 55%.

### 2.2. Modification of Okara Cellulose Crystal with Phenolic Compounds

Modification of the OC with the selected phenolic compounds was performed via a simple impregnation method, as described by Alzate-Arbeláez et al. [6], with some modification. The reaction OC-to-phenolic acid ratios were 1:1, 1:2.5 and 1:5 (*w*/*w*). Initially, the phenolic acids were dissolved in ethanol at a ratio of 1:20 (*w*/*v*) before being divided equally into 3 portions. The OC was mixed with the first portion of the phenolic acid solution at an ambient temperature for 30 min, before drying in a vacuum oven (Memmert; Schwabach, Germany) at 45 °C for 3 h. Then, the mixture was further reacted with the second portion of the phenolic acid solution as with the aforementioned procedure. The process was performed trice to react OC with the entire portions of the phenolic acid solution. Then, the modified OC samples were washed several times with ethanol to remove unreacted phenolic acid, before freeze drying, and analyzing.

#### Characterization of the Modified OC

Particle size: The modified OC samples were dispersed in phosphate buffer (10 mM, pH 7.0) at a concentration of 0.1%. The hydrodynamic size of the particles was determined using a static light particle size analyzer (Zetasizer model nano series; Malvern; Worcestershire, UK) at 25 °C. The volume-weighted diameter (d_4,3_ = ∑ n_i_d_i_^4^/n_i_d_i_^3^) was reported, where n_i_ is the number of particles with diameter d_i_.Structural characteristics: The structural characteristics of the OC as affected by the modification were determined using Fourier transform infrared spectrometry (FTIR; Tensor 27 spectrometer; Bruker; Ettlingen, Germany) with 16 scans, at 4 cm^−1^ resolution over the wavelength range 4000–500 cm^−1^. Moreover, the chemical characteristics of the modified OC were elucidated using 13C solid-state nuclear magnetic resonance (13C NMR; AVANCE II HD/Ascend 400WB; Bruker; Ettlingen, Germany) at 214 MHz with 4000 scans and a spinning rate of 15 KHz.Contact angle: Pellets of cellulosic materials with a planar surface were dropped with 5 μL of distilled water. Then, the contact angles were determined using a contact angle analyzer (OCA-15EC; Dataphysics GmbH; Filderstadt, Germany). The shape and edge characteristic of the water drop on the OC pellet was recorded by a high-speed video camera accompanied by the software of the apparatus. The contact angles formed by both ends of the water drop were then calculated.Phenolic loading efficiency (PE): The cellulose samples were dissolved in ethanol to determine the total phenolic content (TPC) using the Folin–Ciocalteu assay as per the method of Javanmardi et al. [10]. TPC was quantified using gallic acid as a standard, with the PE calculated using Equation (1).
(1)$$PE(\%)=\frac{{TPC}_{complex}-{TPC}_{OC}}{W_{OC}}\times 100$$
where TPC_complex_ and TPC_OC_ represent the TPC values of the modified OC and unmodified OC, respectively, and W_OC_ is the weight of the OC.Antioxidant activity: The DPPH and ABTS radical inhibition capacities were determined as per the method of Thaipong et al. [11], and reported as milli moles Trolox equivalent per gram of sample.

### 2.3. Effect of Modified Okara Cellulose Crystal on Emulsion Stability

The OC and modified OC were dissolved (0.1%, in 10 mM phosphate buffer solution, pH 7.0) before homogenizing (12,000 rpm for 3 min; IKA-T25, Ika Instrument Ltd.; Staufen, Germany) with soybean oil to prepare the emulsion with a 0.1 oil fraction. The emulsions were stored in a screw-cap bottle at an ambient temperature for a period of 2 weeks. Then, the stability of the emulsion was periodically determined during storage.

Colloidal stability: Dispersibility of the emulsions was evaluated by measuring the d_4,3_ of the oil droplets.Oxidative stability: The oxidative degree of the emulsions was quantified by measuring the peroxide value (PV) and the content of thiobarbituric acid reactive substances (TBARs) as per the methods of Prabsangob and Benjakul [12]. PV and TBARs were reported as milligrams hydroperoxide equivalent per liter and milligrams malondialdehyde (MDA) equivalent per liter, respectively.

### 2.4. In Vitro Digestion of the Emulsion

In vitro digestion of the emulsion samples was performed following the method of Le et al. [13] with some modification (see Appendix A). First, the stability of the OC–phenolic complexes was elucidated by determining the TPC and DPPH radical scavenging capabilities after exposure to the gastric and intestinal conditions. Then, the in vitro digestion of the emulsions stabilized using the OC and modified OC was observed. After digestion, the result from the intestinal phase was diluted with DI water and titrated with NaOH standard solution (0.01 mol/L); then, the content of free fatty acids (FFAs) was calculated using Equation (2) [2].
(2)$$FFA(\mu mol)=V_{NaOH}\times C_{NaOH}$$
where V_NaOH_ and C_NaOH_ are the end point volume and concentration, respectively, of the standard NaOH solution. For comparison, the typical emulsion stabilized by the protein, namely bovine serum albumin (BSA), at a corresponding concentration with the cellulosic material was prepared to observe the in vitro digestion in parallel with the studied samples.

### 2.5. Statistical Analysis

Samples were prepared separately in duplicate, with all observed parameters measured in triplicate. The mean values with standard deviations were reported, with the statistical difference between the means determined using analysis of variance and Duncan’s test at the *p* ≤ 0.05 confidence level (SPSS version 12 for Windows; SPSS Inc.; Chicago, IL, USA).

## 3. Results

### 3.1. Characteristics of OC Modified by Phenolic Compounds

The particle size, PE, and antioxidant ability values of the OC and the OC reacted with gallic acid and tannic acid at different ratios are represented in Figure 1. Upon modification, the particle size of the OC tended to increase, while there was no difference in particle size of the OC reacted with gallic acid and tannic acid at a comparable reaction ratio. Increase the phenolic reaction content led to significantly improved PE, and the highest PE were found for the OC–gallic acid (1:5) and OC–tannic acid (1:2.5 and 1:5). A positive correlation has been reported between the phenolic reaction content and PE for the modification of other polysaccharides, including apple cell walls [14] and cellulose nanocrystals [6]. Considering the phenolic acids, a significantly higher PE was observed when the OC was reacted with gallic acid than tannic acid. At the ratio of 1:5, the highest PE values were ca. 70% (ca. 3.8 mM or 646.7 mg GAE/g) and 50% (ca. 3.2 mM or 544.6 mg GAE/g) for the OC–gallic acid and OC–tannic acid complexes, respectively. Different PE values of the modified polysaccharides have been reported, such as 50 mg of berry phenolic extract/g of nanocellulose [6] and 600 mg purified procyanidin/g of apple cell walls [15]. This might be expected due to the dissimilar characteristics of the reacting cellulosic materials and phenolic compounds. The molecular size of the phenolic compounds crucially affected their adsorption capacity onto biopolymers [16,17]. Gallic acid has a low molecular weight with a single phenolic group/mole, whereas tannic acid has a high molecular weight, consisting of 10 phenolic rings/mole [17]. With the larger molecular size, the interaction between tannic acid and the OC might be restricted due to a steric hindrance effect. This effect might be more pronounced especially when the reacting ratio of tannic acid was increased; thus, resulting in the higher PE of the OC–tannic acid compared to OC–gallic acid at the reaction ratio of 1:1. Higher loading percentages onto the polysaccharides of apple cell walls were reported for phenolic compounds with a lower molecular weight compared to those with higher weights [14]. Less cross-linking was reported for the gluten–tannic acid compared to the gluten–gallic acid interaction, which could be explained by the suitable fit between the biopolymer and gallic acid with the smaller molecular size [17]. With the greater presence of hydroxyl residues, interactions between tannic acid and OC might be limited because of electrostatic repulsive forces generated by the sulfate residues of OC and the hydroxyl groups of tannic acid. Electrostatic repulsive force led to restricted phenolic compound binding to pectin, because of the negatively charged uronic acid composition of the pectin [18].

The currently reported free radical scavenging capabilities of the OC could be improved by modification with the phenolics. Generally, significantly greater free radical scavenging ability resulted when the OC was reacted with gallic acid than with tannic acid. The highest DPPH and ABTS radical scavenging abilities were found for the OC–gallic acid (1:5) and OC–tannic acid (1:2.5 and 1:5), which was related to the highest PE of the modified OC, as previously shown. Phenolic compounds exert antioxidant activity via several modes of action, particularly free radical scavenging ability [5]. Improvement of the antioxidant activity of microcrystalline cellulose could be accomplished via grafting with tea phenolics [19].

Next, the FTIR profiles of the OC, as affected by reacting with the phenolic acids, are depicted in Figure 2. Typical FTIR bands corresponding to cellulose were observed for the OC involving the peaks in the range 1400–1300 cm^−1^ relating to a bending of CH and C-OH groups, C=C stretching and/or CH_2_ symmetric bending; the peaks in the range 1150–920 cm^−1^ relating to C-H bonds of CH_2_ groups and stretching of C-O-C in the heterocyclic ring; and the peak at ca. 1020 cm^−1^ associated with C-O stretching and C-H vibration of pyranose rings [6]. The peaks for the gallic and tannic acids typically related to the phenolic compounds involving the spectra at 3100–3500 cm^−1^ relating to C=O stretching [20] and the peak at ca. 1600 cm^−1^ relating to stretching vibration of C=C [21]. There was a noticeable alteration in the FTIR spectra of the OC after the reaction with the phenolic compounds. Peaks emerged at 3100–3500 cm^−1^ and 1600 cm^−1^ for the OC that reacted with gallic acid (1:2.5 and 1:5) and tannic acid (1:1, 1:2.5, and 1:5) implying the integration of the phenolic compounds to the OC structure. Cross-linking between the sesame phenolic extracts and carboxymethyl cellulose was also confirmed via another FTIR study [9]. The chemical modification of cellulose particles by interacting with phenolic compounds could have occurred at the hydroxyl residues, mainly via non-covalent interactions involving hydrogen and hydrophobic bonding, as well as ionic interaction [14].

The ^13^C NMR spectra of the OC interacted with gallic acid (1:5) and tannic acid (1:2.5) providing the highest PE were elucidated to confirm the surface modification of OC using the phenolic compounds, as shown in Figure 2C. Typical ^13^C NMR spectra of cellulose were found for the unmodified OC, including the peaks at *δ* 107–105 ppm corresponding to the C-1 of α-D-glucopyranose ring, *δ* 85–90 ppm relating to the C-4 of the crystalline and amorphous regions of cellulose, and *δ* 70 ppm indicating C-2, C-3 and C-5 of the pyranose rings [22]. After interacting with the phenolic compounds, the NMR spectra of the OC were clearly affected, suggesting the successful inclusion of the phenolics in the OC structure. For the modified OC, spectra relating to the phenolic residue were present, involving the peaks around the *δ* 110 ppm (C-2′, C-6′), *δ* 122 ppm (C-1′), *δ* 137 ppm (C-4′), *δ* 145 ppm (C-3′, C-5′), and *δ* 170 ppm (C-7′) [23]. In addition, the intensity of the peaks relating to the cellulose decreased, implying the coating of the phenolics on the OC particles and, thereby, swamping the NMR signal of the cellulose [24].

The highest antioxidant ability levels for the OC modified by gallic acid (1:5) and tannic acid (1:2.5) at the selected ratio were used for a further study and named as OC-G and OC-T, respectively.

### 3.2. Effect of Modified OC on the Physical and Chemical Stability of Emulsion

The surface properties and emulsifying ability of the OC and modified OC were studied. Figure 3A shows the contact angles of the cellulosic materials. Notably, the modification using dissimilar phenolic compounds affected the surface property of the OC in different ways. The OC had a contact angle of ca. 87.5°, suggesting their equal wettability in the oil and water phases. This led to the emulsion-stabilizing effect of the OC due to their anchoring ability to the interfacial areas between oil and water [25]. Improved hydrophilicity of the OC could be accomplished by inclusion with gallic acid, as evident by the lowered contact angle of the OC-G compared to the unmodified OC. This behavior might be supposed since the OH residuals of the gallic acids grafted on the OC surfaces. The OC-G with a contact angle of ca. 74.4° is partially hydrophilic, implying its favorable distribution in the aqueous phase of the emulsion. Particles with hydrophilicity could form O/W Pickering emulsion effectively due to their affinity to the eternal phase of the system [26]. On the other hand, modification using tannic acid led to an increased contact angle of the OC, implying improved hydrophobicity of the particles. The current contradiction behavior of the contact angle of the OC after modification using gallic acid and tannic acid was presented which might be explained due to different molecular characteristics of the selected phenolic compounds. Increased hydrophobicity of the OC-T might be expected due to the larger molecular size of tannic acid than for gallic acid. The positive correlation between the molecular size and hydrophobicity of the phenolic compounds has been confirmed [27]. Incremental surface hydrophobicity was reported in other studies using tannic acid to modify cellulose nanocrystals [8] and to modify microcrystalline cellulose [19]. The increased surface hydrophobicity of the cellulosic materials by inclusion with tannic acid led to improved surface activity [28] and emulsifying capability of the modified cellulose [8,19].

From Figure 3B, better colloidal stability could be found for the emulsions stabilized by the modified OC compared to the native OC, as indicated by the smaller droplet size increasing with the storage time of the OC-G- and OC-T-based emulsions. This behavior might be explained by the alteration of the surface properties of the OC after modification, as previously suggested. Superior dispersibility of the emulsions stabilized using OC-G and OC-T than from using OC might have occurred due to the increased particle size of the of OC after modification, as previously suggested in Figure 1A. An emulsifier with a larger particle size could form interfacial films with increased thickness, thereby facilitating emulsion dispersibility via an effective steric stabilization mechanism [3,19,29].

Oxidative stability of the emulsions was determined by measuring storage time dependence on the quantity of PV (Figure 3C) and TBARs (Figure 3D). For the OC- and OC-T-based emulsions, the PV values increased initially before declining. A diminishing PV could have been due to degradation of peroxides and/or the transformation of the compounds to secondary oxidative products, such as ketones and aldehydes [12]. However, for the OC-G-based emulsion, the PV increased continuously throughout the storage, implying less development of secondary oxidative products. Furthermore, the OC-G-based emulsion had a lower TBARs increment than the counterparts stabilized using OC and OC-T. These behaviors suggested better oxidative stability of the emulsion stabilized using OC-G than for the native OC and OC-T, which might have been due to the higher PE and better antioxidant activity of the OC-G, as previously suggested in Figure 1. Moreover, in the current study, perhaps the OC-G had the highest hydrophilicity compared to the OC-T. Greater oxidative susceptibility was reported for a Pickering emulsion stabilized using the emulsifier with higher hydrophobicity [29,30]. Upon modification with phenolic compounds, the cellulose crystals could promote effective oxidative stability of the salad dressing [6]. The mechanism of lipid oxidation in a multiphasic system is complicated, with the oxidative rate depending on several factors, particularly the partitioning of the emulsifier and/or antioxidant agent in different phases of the emulsion [31].

### 3.3. In Vitro Digestion of Emulsions Stabilized Using Modified OC

Figure 4 shows the TPC and DPPH radical scavenging abilities of the OC and the modified OC after exposure to gastric and intestinal digestion. The modified OC had good stability against the harsh acidic conditions of the gastric phase, as suggested by no change in the TPC or the free radical scavenging ability of the OC-G and OC-T until the end of gastric digestion. Stability of the phenolic compounds present in black carrots involved anthocyanins, ferulic acid and caffeic acid, against gastric digestion has been suggested [32]. The good stability of the modified OC in the gastric phase might have allowed for the transport of the phenolics to the lower GIT where they could further exert beneficial effects due to their availability for intestinal absorption [7] and gut microbial metabolism in the large intestine [16]. The susceptibility of bioactive compounds through the upper GIT severely limited bioaccessibility of the compounds, because of the biotransformation and transportation of phenolics having mostly taken place in the intestine and hepatic tissues via xenobiotic metabolism [7].

Considering the intestinal phase, the TPC of the modified OC continuously decreased with digestion time. Although the OC-G had a higher TPC than the OC-T, the TPCs between the two samples were comparable from the second hour entirely due to intestinal digestion. Interaction between the phenolics and cellulose was expected to be mainly non-covalent and regarded as a weak bond [14]. Therefore, withdrawing the phenolic acids from the surfaces of the OC could be expected after exposure to digestive enzymes [6]. In the final stage of the intestinal phase, the TPC availability levels were ca. 3.8 mM GAE/g and 3.2 mM GAE/g corresponding to 18.4% and 12.5% of phenolic displacement for the OC-G and OC-T, respectively. The diminishing level of TPC was concomitant with the lowering DPPH radical scavenging ability of the modified OC after exposure to intestinal digestion. This behavior might have been due to destruction of the phenolic compounds in the intestinal phase due to the occurrence of several chemical reactions, such as partial oxidation and polymerization [29,32]. Considering the OC, the changes on TPC and DPPH radical scavenging ability were not observed entirely through gastric and intestinal phases implying less effects of the simulated gastric and intestinal digestion on the property of the OC without phenolic acid grafting. This might be supposed due to indigestibility of cellulosic materials in the upper GIT [33].

Next, the emulsions stabilized by the native and modified OC were subjected to in vitro digestion; the FFA contents liberated from the emulsions are quantified in Figure 5. Traditional emulsion is always stabilized by proteins, so the FFA content liberated from the BSA-based emulsion was also observed for comparison. The increased oil content of the emulsions led to significantly higher FFA liberation from the emulsion. There were no significant differences in the amounts of FFA released from the emulsions stabilized using OC, OC-G, or OC-T at both the observed oil fractions. For the emulsions with the 0.1 oil fraction, the FFA was significantly lower for the emulsions stabilized using the cellulosic materials compared to the counterparts stabilized using BSA, suggesting the lipolysis retardation ability of the OC and modified OC. This result was in accordance with the reports of DeLoid et al. [34] and Liu et al. [35]. Cellulose exerted a retardation effect on the digestibility and absorbability of triacylglycerols (TAG) through a non-specific binding with the other components present in the human GIT, particularly the digestive enzymes, bile salts, and lipid digestive products [4]. Furthermore, a higher lipolysis degree of the BSA-based emulsions might be postulated due to the collapse of the emulsion after exposure to the gastric phase, due to the harsh acidic conditions cleaving protein molecules [36]. Therefore, the emulsified oil droplets were exposed and easily attacked by the lipase and bile salts in the intestine, resulting in a greater amount of liberated FFA. On the other hand, for the emulsions stabilized using the native and modified OC, the interfacial film of the cellulosic materials could provide steric hindrance to inhibit lipase activity, thereby lowering the degree of lipolysis [34]. Notably, the emulsion stabilizing effect of the OC was mainly expected due to steric barrier formation [3].

For the emulsions with the 0.2 oil fraction, the effect of the OC and modified OC on retarding lipolysis was not significant, as indicated by their comparable released FFA contents with that from the BSA-based emulsion. In the current study, the emulsifiers were applied at a fixed concentration. Larger interfacial areas could be supposed for emulsions with a higher oil content, so bare interfacial areas might be available for the emulsions with a higher oil fraction. This might lead to less ability of the OC and modified OC to render lipolysis due to the presence of intact areas between TAG and the digestive enzymes, resulting in increased FFA content. The presence of the interfacial gaps allowed for accessibility of the lipase and bile salts, thereby facilitating digestion of the TAG at oil droplet surfaces [37]. The current results emphasized the importance of the steric barrier provided by the cellulosic materials in the protection of the emulsified oil droplets from lipolysis.

## 4. Conclusions

Surface modification affected characteristics of the OC and led to improved emulsifying and antioxidant capabilities of the modified OC successfully. Modification of the OC using gallic acid led to a higher PE and greater antioxidant activity than from using tannic acid. The emulsifying capability of the modified OC was developed as suggested by the improved colloidal stability and oxidative stability of the Pickering emulsion. As a result of the modification by grafting with phenolic residues, particle size of the OC increased that might lead to improved dispersibility and lowered lipolysis of the emulsion via a steric barrier formation. The modified OC had good stability against the gastric phase of a simulated GIT, suggesting the possibility of the modified OCs exerting beneficial health effects due to their availability in the lower part of the GIT. The native and modified OC exhibited a retardation effect on the in vitro lipolysis degree of the emulsion and a lower liberated FFA content was found compared to the protein-based emulsion. The current work suggested the feasibility of the modified OC as an innovative emulsifier to promote both colloidal and oxidative stability of the emulsion. In addition, the interfacial layers of the modified OC could lower lipolysis of fat-containing food that might be useful for the development of food to counteract obesity. However, the cytotoxicity of the modified OCs should be further studied to ensure their safe utilization in food products. Furthermore, elucidation of the kinetics of phenolic loading onto the OC particles and of the lipolysis retardation mechanism of the modified OCs may facilitate the understanding of the characteristics and functional properties of the modified OCs.

## Figures and Tables

**Figure 1 foods-13-00184-f001:**
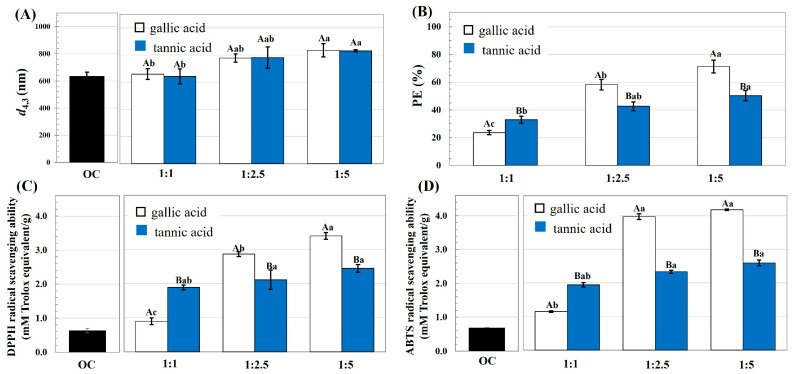
(**A**) Particle size, (**B**) PE, (**C**) DPPH radical scavenging ability, and (**D**) ABTS radical scavenging ability of OC (■) and OC reacted with gallic acid (□) or tannic acid (■) at different ratios of 1:1, 1:2.5 and 1:5. In each subfigure, different lowercase letters indicate significant (*p* ≤ 0.05) differences between the means as affected by the OC-to-phenolic reaction ratio, and different capital letters indicate significant (*p* ≤ 0.05) differences between the means as affected by the phenolic type.

**Figure 2 foods-13-00184-f002:**
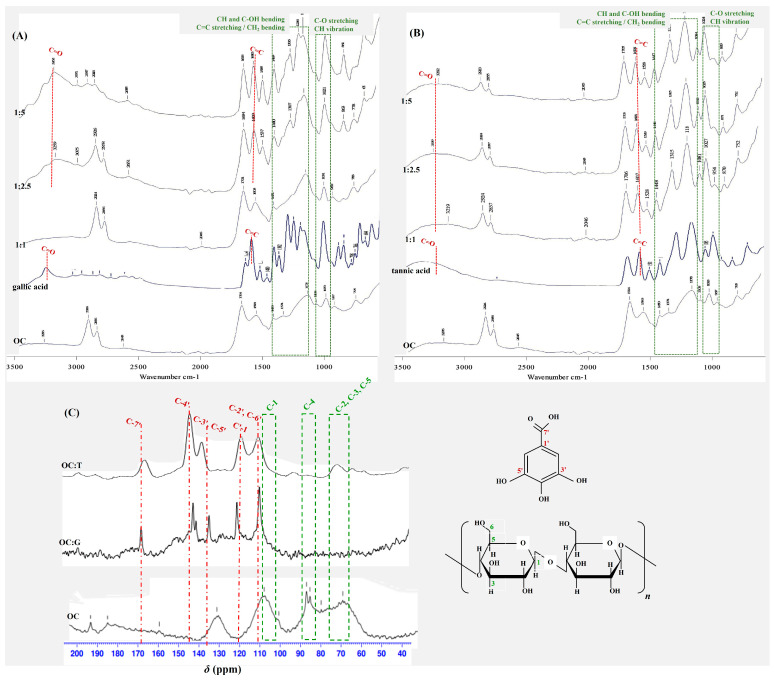
FTIR profiles of OC modified with (**A**) gallic acid or (**B**) tannic acid at different ratios (1:1, 1:2.5 and 1:5) and (**C**) ^13^C-NMR spectra of OC and OC modified with gallic acid (1:5) or tannic acid (1:2.5) at the selected ratio.

**Figure 3 foods-13-00184-f003:**
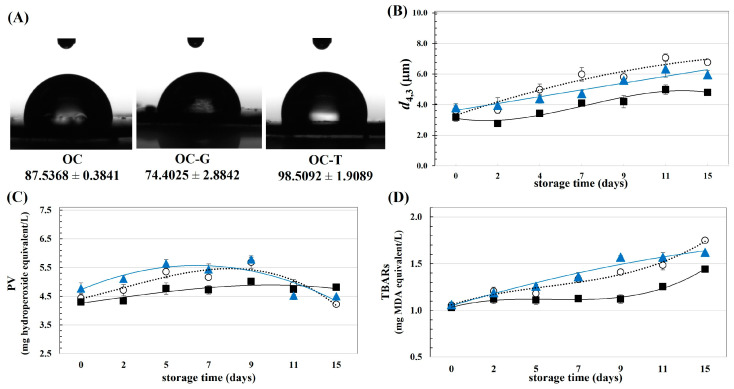
(**A**) Contact angles of cellulosic materials, (**B**) oil droplet size, (**C**) PV and (**D**) TBARs content of emulsions stabilized using OC (◯), OC-G (■) or OC-T (▲) as a function of storage time.

**Figure 4 foods-13-00184-f004:**
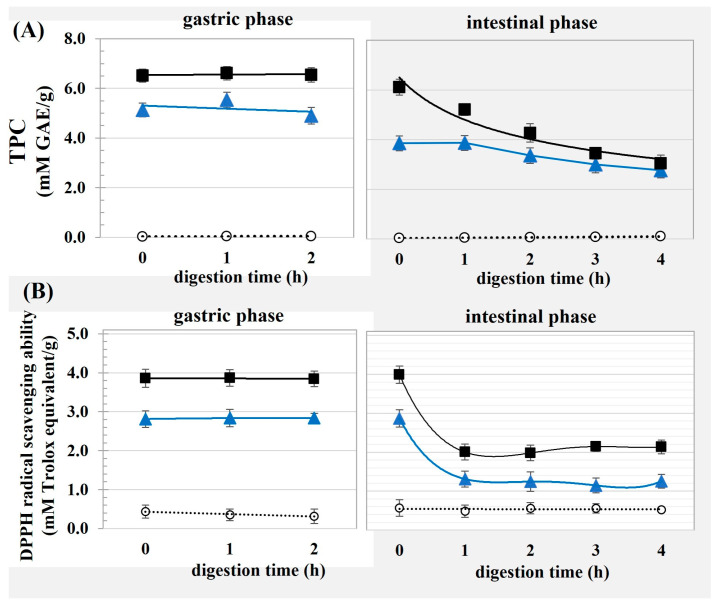
(**A**) TPC and (**B**) DPPH radical scavenging ability of OC (◯), OC-G (■) or OC-T (▲) after exposure to in vitro gastric and intestinal phase digestion at different times.

**Figure 5 foods-13-00184-f005:**
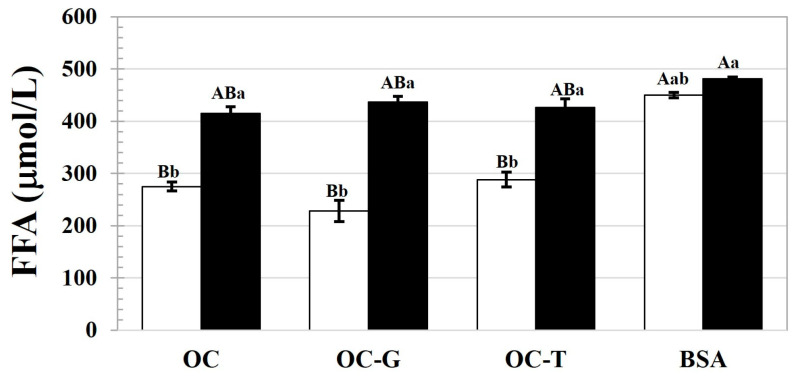
FFA content liberated from emulsions stabilized using OC, OC-G, OC-T or BSA for emulsions containing oil fractions of 0.1 (□) and 0.2 (■). Different lowercase letters indicate significant (*p* ≤ 0.05) differences between the means as affected by the oil fraction, and different capital letters indicate significant (*p* ≤ 0.05) differences between the means as affected by the emulsifier type.

## Data Availability

Data is contained within the article and Appendix A.

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
