# Peer review of "Surface Modification of Okara Cellulose Crystals with Phenolic Acids to Prepare Multifunction Emulsifier with Antioxidant Capacity and Lipolysis Retardation Effect"

_foods, 2024, doi:10.3390/foods13020184_

Round 1

Reviewer 1 Report

Comments and Suggestions for Authors

1.       L19... present the value for LE for gallic acid and tannic acid

2.       This statement is inappropriate in this part of abstract… Surface modification based on grafting with phenolic acids could improve capability of the OC to enhance both the colloidal and oxidative stability of the emulsion. Why use “could” when you already have the results, rather make a statement based on the observed data.

3.       I suggest that the latter part of the abstract be thoroughly rewritten, highlighting and making specific inferences based on the data obtained in this study

4.       L96 (preparation of OC) was not well described. Treatment temp, time and other conditions are missing. Rewrite in detail.

5.       L104 0.75 m?? check this again

6.       L 110. How was OC dissolved before mixing?

7.       Please arrange modified OC characteristics under a sub-section (2.2.1).

8.       The figure’s quality is so poor. Must be improved Fig 1, 2, 3, & 4

9.       The conclusion statement lacks scientific flow. The first sentence must be rephrased.
- A successful modification and characterization of OC preceded the its application as an emulsifier.

10.   Different types of phenolic compounds influenced the surface properties of the OC in different ways… (only 2 types were used, and in some instances, the observed effects were not too obvious. So, tone down this statement)

Comments on the Quality of English Language

Minor errors should be avoided

Author Response

please find the response as the attached file. 

Reviewer 2 Report

Comments and Suggestions for Authors

The article provides a detailed study on the changes in characteristics (particle size, phenolic loading, antioxidant activity, chemical structure, contact angle, oxidation value, gastrointestinal digestibility) of okara cellulose crystals modified with two typical phenolic acids, providing a theoretical basis for their performance improvement strategies as emulsifiers. However, there are some small issues that need to be raised. Keywords: Sorting keywords in the order they appear in the article will make them more organized. 

Line 104: The diameter unit of OC is incorrect.

Line 131: The operation method and details of the contact angle should be explained to make the method more valuable for reference. 

Line 141: The description of wt is incorrect, and similar errors have also occurred elsewhere. 

Line 361: Explaining the reasons why the PV value of OC-T is different from other substances will make the research content more complete. 

Conclusions: A more detailed summary of the research results should be provided, such as the particle size changes and emulsification performance after phenolic acid modification.

Author Response

(The authors gave the same response as above.)

Round 2

Reviewer 1 Report

Comments and Suggestions for Authors

Required modifications were satisfactorily done

Comments on the Quality of English Language

Please make thorough grammatical editing